# Mpox Incidence and Vaccine Uptake in Men Who Have Sex with Men and Are Living with HIV in Denmark

**DOI:** 10.3390/vaccines11071167

**Published:** 2023-06-27

**Authors:** Anne-Sophie Winther Svartstein, Andreas Dehlbæk Knudsen, Safura-Luise Heidari, Line Dam Heftdal, Marco Gelpi, Thomas Benfield, Susanne Dam Nielsen

**Affiliations:** 1Department of Infectious Diseases, Rigshospitalet, University of Copenhagen, 2100 Copenhagen, Denmark; asvartstein@gmail.com (A.-S.W.S.); andreas.dehlbaek.knudsen@regionh.dk (A.D.K.); line.dam.heftdal.01@regionh.dk (L.D.H.);; 2Department of Cardiology, The Heart Center, Rigshospitalet, University of Copenhagen, 2100 Copenhagen, Denmark; 3Department of Infectious Diseases, Copenhagen University Hospital—Amager and Hvidovre, 2650 Hvidovre, Denmark

**Keywords:** mpox, HIV, vaccination, men who have sex with men, vaccination willingness

## Abstract

(1) Background: Here, we investigate the incidence of mpox and factors associated with vaccine uptake in mainly well-treated men who have sex with men and are living with HIV (MSMWH). (2) Methods: This study included 727 MSMWH from the Copenhagen co-morbidity in HIV infection (COCOMO) study from 1 May to 31 October 2022. Mpox infection and vaccination status were obtained from the Danish Microbiology Database and The Danish Vaccination Register. Vaccination willingness was assessed through an online survey. (3) Results: At a median follow-up of 180 days, 13 (1.8%) participants had laboratory-confirmed mpox infections. Furthermore, 238 (32.7%) had received the mpox vaccine. A sexually transmitted disease (STD) in the preceding two years was associated with a higher risk of mpox infection (hazard ratio 7.1; 95% confidence interval (CI) [1.9–26.9]) and with higher odds of vaccination (adjusted odds ratio 3.1; 95% CI [2.2–4.6]). 401 (55.2%) participants responded to the survey. 228 (57.0%) reported very high vaccination willingness. The self-perceived risk of infection was associated with vaccine uptake. (4) Conclusions: The incidence of mpox was low. A prior STD was associated with both a higher risk of mpox infection and higher odds of vaccination. Despite high-risk sexual behavior and high vaccination willingness, a sizable fraction of participants had not been vaccinated.

## 1. Introduction

Since the eradication of the smallpox virus, mpox is considered to be the most important orthopoxvirus infection that affects humans [1]. The clinical presentation of the disease is often a systemic illness with influenza-like symptoms including headache, fatigue, muscle aches, fever, and lymphadenopathy. Subsequently, some patients develop a vesiculopustular rash, which can be painful or itchy [2], whereas others experience only a few or no symptoms [3]. Mpox is endemic in African regions [4]. However, during May 2022, a rapidly emerging outbreak of mpox in Denmark and many other non-endemic countries raised concern. As of June 13, 87,979 cases of mpox and 147 deaths due to mpox had been reported, corresponding to a fatality rate of 0.17%. As a comparison, the current global case fatality rate for COVID-19 is in the range of 0.05–0.5% [5,6].

Men who have sex with men (MSM) and people living with HIV (PLWH) have been identified as risk groups for mpox infection [6]. The Centers for Disease Control and Prevention (CDC) have estimated that among people diagnosed with mpox in the United States, approximately 40% had HIV [7]. Clinical data on mpox in men who have sex with men and are living with HIV (MSMWH) and are mainly well-treated are limited, and it is relevant to investigate the incidence of mpox among this population in order to characterize the pandemic.

Mpox is preventable with a live, non-replicating smallpox and mpox vaccine that provides approximately 78% protection against mpox 14 days after vaccination [8]. In Denmark, persons at high risk of mpox infection have been prioritized for vaccination, and, accordingly, MSMWH are eligible for vaccination [9]. Most existing studies have focused on the vaccination willingness of either MSM, health care workers, or the general population. Less is known about vaccine willingness in MSMWH, who are considered a high-risk population. Knowledge of vaccination uptake and identification of influencing factors in a specific high-risk population are crucial for the development of targeted intervention strategies and public health efforts in order to mitigate the impact of mpox.

This study aimed to investigate (1) the incidence of mpox, (2) the uptake of the mpox vaccine, and (3) factors associated with vaccine uptake in a cohort of mainly well-treated MSMWH.

## 2. Materials and Methods

### 2.1. Study Design

Participants were recruited from the Copenhagen Comorbidity in HIV Infection (COCOMO) study. The COCOMO study is an observational, longitudinal study that aims to investigate the burden and pathogenesis of non-AIDS comorbidity in PLWH in Copenhagen. The study enrolled 1099 participants between March 2015 and November 2016, corresponding to more than 40% of PLWH in the greater area of Copenhagen. Details concerning recruitment and data collection have previously been described [10]. Inclusion criteria for the present study were MSM > 18 years of age, with HIV-1 infection, alive, and resident in Denmark on 1 May 2022. A supplementary online survey was sent out to all participants.

### 2.2. Vaccination

On 23 May 2022, the first case of mpox was observed in Denmark, which led to the initiation of a vaccination strategy by the Danish health authorities to prevent mpox infection in those at high risk. The live, non-replicating smallpox and mpox vaccine (Bavarian Nordic) is approved by the European Medicines Agency (EMA) [11]. The vaccination program commenced on 10 August 2022.

People at high risk of mpox included those receiving pre-exposure prophylaxis for HIV (PrEP), those eligible for PrEP, or those who had similar risks, including people receiving antiretroviral treatment [9]. Additionally, close contacts of confirmed cases of mpox were offered vaccination. Persons who had previously been vaccinated against smallpox were offered a single booster dose of the mpox vaccine Imvanex. Persons without a previous smallpox vaccine were offered two doses of Imvanex with an interval of 28 days.

### 2.3. Outcomes

The primary outcomes of this study were:(1)A positive mpox test;(2)Receiving the mpox vaccine.

Outcomes were investigated in all participants, and data was collected between 1 May 2022 and 31 October 2022.

Data on positive mpox test results was obtained from the Danish Microbiology Database (MiBa), a national database with surveillance of infectious diseases across Denmark and complete, national coverage of all microbiological tests since 2011 [12]. MiBa gives healthcare professionals access to microbiological tests and results from both inpatient and outpatient clinics, as well as from general practitioners.

Information on mpox vaccination status was obtained through The Danish Vaccination Register (DDV) [13]. DDV has been a national database with mandatory registration of all administered vaccines since 2015 [13].

### 2.4. Covariates

Information about alcohol, smoking, origin, and education was collected from self-reported questionnaires. Alcohol consumption was defined as the weekly intake of alcohol in grams and furthermore categorized as less or more than 10 units a week according to the Danish Health Authority’s recommendations [14]. Smoking status was categorized as never smoker, current smoker, or ex-smoker. Origin was categorized as Scandinavian, other European, and other. Educational level was defined as education acquired after high school and divided into the following two subcategories: No education or short education (including short (<3 years) and vocational), and long education (including education ≥ 3 years, e.g., nurse or teacher and university degree). Information regarding previous sexually transmitted diseases (STDs), including syphilis, chlamydia, gonorrhea, genital herpes, genital warts, and others, in the period of 23 May 2020–23 May 2022, was obtained from MiBa. Syphilis was included in the definition of an STD and was diagnosed with either a swap PCR (polymerase chain reaction) test for Treponema Pallidum from a lesion, a nontreponemal test (Wassermann reaction (WR) and rapid plasma reagin (RPR)), or with more extended serology tests. Syphilis was defined as documented seroconversion or a fourfold increase in titer of RPR over the last 12 months. MSM status, current plasma HIV viral load (copies/mL), and blood CD4^+^ T lymphocyte count (cells/μL) were obtained from participants’ health records.

### 2.5. Survey

To collect information regarding vaccination willingness, vaccination hesitancy, and mpox risk perception, an online survey was conducted among the included participants. The survey was sent to all and was responded to by 401 of the participants. Participants were invited through Digital Post, a secure, personal mailbox that allows for secure digital communication between public authorities and Danish residents. Vaccination willingness was measured on a Likert scale of 1 to 5 (1 = very low willingness and 5 = very high willingness) (see Appendix A). Participants were also asked to report the number of sexual partners in the last 12 weeks and whether they had received the smallpox vaccine as a child.

An electronic reminder was sent to all participants.

### 2.6. Statistical Analysis

Continuous variables were compared using the Mann-Whitney U test or the student’s t-test. Normally distributed variables were reported as means with standard deviations (SD), while non-normal deviations were reported as medians with interquartile ranges (IQR). Categorical variables were compared using the chi-squared test or Fisher’s exact test and were reported as numbers and percentages (%).

To identify factors associated with vaccination uptake, univariable and multivariable logistic regressions were performed. All factors with a *p*-value < 0.1 in the univariable analysis were retained in the multivariable models. The multivariable models included age, a diagnosis of at least one STD in the preceding two years, and smoking status. The cumulative incidence of mpox was calculated considering the competing risk of dying using the Aalen-Johansen estimator.

To assess whether various variables could influence the outcome of mpox infection, Cox proportional hazards regression models were used. Model 1 was adjusted for age and history with at least one STD in the preceding two years.

To further investigate the association between a previous STD diagnosis and vaccine uptake, a sensitivity analysis including only STDs in the preceding 6 months was carried out.

For all analyses, a *p*-value ≤ 0.05 was considered statistically significant. All statistical analyses were conducted in R V.3.4.2 (R Foundation for Statistical Computing).

## 3. Results

### 3.1. Baseline Characteristics

Of the 1099 participants in the COCOMO study, 49 had died and 21 had emigrated before 1 May 2022. Of the remaining 1029 participants, 727 were MSM and included in the present study. Baseline characteristics are shown in Table 1. The median age of the participants was 55.7 years (IQR: 48.3–62.8).

Missing data for each covariate and for survey responses are shown in Appendix A.

### 3.2. Incidence of Mpox Infection and Related Risk Factors

Of the 727 participants, the cumulative incidence of laboratory-confirmed mpox infection during the six months of follow-up was 13 (1.8%), equivalent to 35.8 cases per 1000 person-years. The median age among cases was 47 years (IQR: 43–53). Among participants with an mpox infection, 10 (76.9%) were diagnosed with at least one STD in the preceding two years, compared to 174 (24.4%) of the participants without an mpox infection. Of the 13 participants who were infected with mpox, 11 had a laboratory-confirmed mpox diagnosis before the vaccination campaign rollout commenced.

In univariable Cox models, a diagnosis with at least one STD in the preceding two years and younger age were significant predictors of mpox infection (Appendix A). After adjustment for age and a history of STDs, only a diagnosis of one or more STDs in the preceding two years remained a significant independent predictor of mpox infection [hazard ratio (HR) 7.1; 95% CI 1.9–26.9].

### 3.3. Determinants Associated with Vaccine Uptake

A total of 238 participants (32.7%) received at least one dose of the mpox vaccine during the follow-up period. Participant characteristics, stratified by mpox vaccination status, are shown in Table 2.

In univariable analyses, a diagnosis with at least one STD in the preceding two years was associated with higher odds of vaccine uptake (Appendix A). Higher age and being a current smoker were associated with lower odds of vaccination uptake, whereas there were no statistically significant associations between alcohol intake, origin, educational length, undetectable viral load, or CD4^+^ cell count and vaccine uptake.

In multivariable logistic regression, a diagnosis with at least one STD in the preceding two years was independently associated with higher odds of vaccine uptake (adjusted odds ratio (aOR) 3.1; 95% CI [2.2–4.6]). The average time from a diagnosis of an STD to study start was 216 days (IQR: 88–418 days). In a sensitivity analysis including only STDs in the preceding 6 months, results were similar. Being a current smoker was associated with lower odds of vaccine uptake (aOR 0.6; 95% CI [0.4–0.9]), and higher age was not associated with vaccine uptake in multivariable analysis.

### 3.4. Survey Results

Of the 727 included participants, 401 responded to the online survey (a response rate of 55.2%). Survey responses are shown in Figure 1. The characteristics of respondents and non-respondents are given in Table 3. There were significant differences in weekly alcohol intake, smoking status, and origin between respondents and non-respondents, but they did not differ significantly in any other characteristics.

Almost all (99.2%) of the respondents had heard about mpox, and 373 (95.6%) had heard about the vaccine. Of all respondents, 171 (44.0%) reported having been offered the mpox vaccine, while 155 (39.7%) reported having received the vaccine. More than half of the respondents (57.0%) answered that they were very willing to receive the vaccine, while 46 (12.7%) of the respondents had a very low willingness towards receiving the vaccine. A total of 221 (55.4%) of the respondents perceived themselves as being at risk of being infected with mpox. Perceiving oneself as being in the risk group for mpox infection was the predominant reason for choosing to get vaccinated, whereas the predominant reason for choosing not to get vaccinated was the perception of not being in the group at risk of mpox infection (Figure 1f,g). Of those who were very willing to receive the vaccination, 35.7% had not yet been offered the vaccine.

Table 4 shows the differences between vaccinated and unvaccinated respondents regarding concern for mpox, number of partners in the last 12 weeks, and STD diagnosis in the preceding two years.

## 4. Discussion

In the present study, we investigated the incidence of mpox, uptake of the mpox vaccine, and associating factors in a cohort of mainly well-treated MSMWH. In 727 participants, we found a cumulative incidence of mpox of 1.8% during six months of follow-up, corresponding to 35.8 cases per 1000 person-years. One in three participants had received the mpox vaccine. A diagnosis with at least one STD in the preceding two years was associated with both a higher risk of becoming infected with mpox and higher odds of mpox vaccine uptake. Furthermore, an online survey highlighted that high-risk sexual behavior and the perception of being at risk of infection were associated with vaccine uptake. Importantly, a notable fraction of high-risk individuals had not received the vaccine.

Of the 13 individuals who were diagnosed with mpox, the majority had been diagnosed with at least one STD in the preceding two years, and having had an STD was associated with a higher risk of mpox after adjusting for age. This is consistent with previous studies on the current outbreak from the US and the UK [15,16], where 41% and 54%, respectively, of individuals infected with mpox reported a prior STD. Moreover, we found a diagnosis with at least one STD in the preceding two years to be associated with higher odds of vaccine uptake, which is also consistent with other studies [17,18]. It has been debated whether mpox should be defined as an STD [19]. The localization of the characteristic rash of mpox to the genitalia and perianal area [20], as well as the high prevalence of a prior STD in individuals who have been diagnosed with mpox, support the theory of mpox being a predominantly sexually transmitted disease.

The current mpox outbreak has been reported to disproportionately occur among MSM [21]. Additionally, individuals with mpox have been found to have a high prevalence of HIV infection [20,22,23]. This underscores the importance of improving efforts to prevent mpox in MSMWH, and vaccination is a mainstay of such efforts. In the present study, vaccine uptake was 32.7%, which is slightly higher than in an American survey [24] and slightly lower than in a Canadian survey [25], where vaccine uptake was reported to be 18.3% and 51.0%, respectively. Of note, we reported vaccine uptake based on data from the national registry, DDV, in which registration of all administered vaccines in Denmark is mandatory. Thus, recall bias, selection bias, and other factors inherent to surveys did not influence our data. Nevertheless, in this study, less than a third of MSMWH were vaccinated. It is important to note, however, that there is a potential for underestimation of the actual vaccine uptake due to data collection taking place during the vaccination period. The participants had had 2.5 months to receive the vaccine when the data was collected.

We found that being a current smoker was associated with lower odds of receiving the vaccine. Consistently, previous research has found that current smokers hold more negative attitudes towards vaccines for other viruses [26,27,28]. Several potential explanations may exist for this phenomenon, including more widespread vaccine skepticism among smokers, a propensity to engage in health-risk behaviors, and unmeasured confounding.

The survey responses showed that the number of sexual partners was higher among vaccinated respondents compared to unvaccinated respondents. Likewise, the majority of those who were diagnosed with at least one STD in the preceding two years had received the mpox vaccine. Given that both the number of sexual partners and prior STD diagnoses are risk factors for mpox, this suggests that individuals at higher risk of acquiring and transmitting mpox are indeed taking steps to prevent transmission. Similar trends have been observed in other studies [24,29].

Over half (57.0%) of the respondents reported being very willing to receive the vaccine, but only 44.0% reported being offered the vaccination. Furthermore, a sizable fraction of high-risk individuals reported not having received the vaccination. This indicates a potential for improvement in terms of educating and informing both healthcare professionals and patients. A recent literature review highlighted the limited knowledge of mpox among health care workers [30]. Thus, educating healthcare professionals on identifying high-risk individuals and providing the knowledge to inform and administer the vaccine against mpox should be considered an important part of the vaccination strategy.

A notable fraction (18.2%) of the respondents reported very low vaccination willingness. Effective communication and continued education among patients are known moderators for increasing vaccine uptake [31] and are thus crucial components in reducing vaccine hesitancy. An important factor in communication about the mpox vaccine is to reduce stigmatization [32]. Stigma-related challenges in accessibility to the mpox vaccine have previously been described [25]. Similar to past infectious disease outbreaks, such as HIV, stigma can significantly impede health and well-being [33].

In this survey, a vast majority of the respondents (98.3%) considered the mpox vaccine to be effective. This may indicate that the information communicated by The Danish health authority and health care professionals regarding the efficacy of the mpox vaccine was effective and sufficient to prevent unsubstantiated vaccine speculations among MSMWH.

One of the strengths of our study is that it focuses solely on MSMWH and aims to identify possible predictors for mpox infection and vaccine uptake in this specific population. Additionally, the study benefits from the use of a well-characterized observational cohort and national databases that are widely recognized for their reliability and validity. The survey added information about the vaccine’s uptake and helped assess the participants’ self-perceived risk and attitudes towards the vaccine. However, there were some limitations to the study. Specifically, only 13 of 727 MSMWH were diagnosed with mpox infection during the study period, which limited the power to detect possible predictors of infection. Confounding from missing data may also have influenced our results. The generalizability of this study is limited. While our findings provide insights into the mpox incidence and vaccine attitudes of mainly well-treated MSMWH, they may not be applicable to MSMWH who have not received adequate treatment or are immunosuppressed, nor to individuals who are living in settings without access to universal health care free of charge. Moreover, information about alcohol consumption, smoking, origin, and education was collected through self-reported questionnaires and is subject to recall bias and social desirability bias. However, data were collected before the mpox outbreak. Additionally, the online survey data may not be fully representative of all study participants, as there were significant differences between respondents and non-respondents.

## 5. Conclusions

In conclusion, in a cohort of mainly well-treated MSMWH, the incidence of laboratory-confirmed mpox infection was low, but only one in three had received the mpox vaccine. Despite high-risk sexual behavior and high vaccination willingness, a sizable fraction of participants had not been vaccinated, and some of these had not been offered the vaccine.

Our findings suggest the need to improve the education of both healthcare professionals and patients when planning vaccination strategies among MSMWH.

## Figures and Tables

**Figure 1 vaccines-11-01167-f001:**
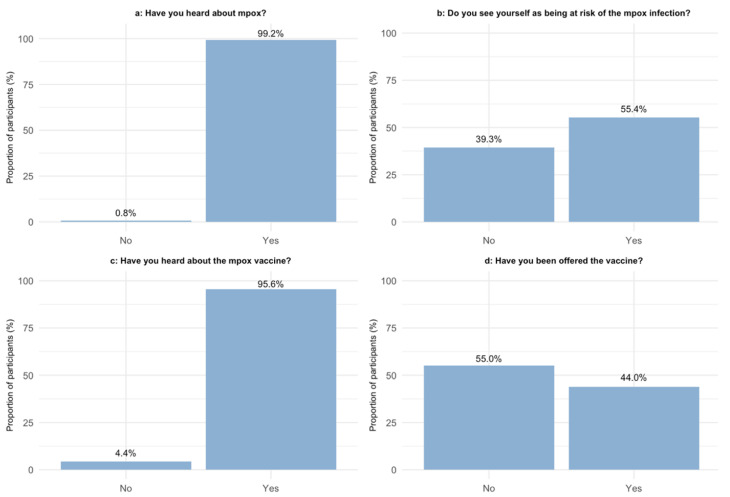
Awareness and attitude towards mpox and mpox vaccination among MSMWH: (**a**) Question: “Have you heard about mpox?”; (**b**) Question: “Do you see yourself as being at risk of the mpox infection?”; (**c**) Question: “Have you heard about the mpox vaccine?”; (**d**) Question: “Have you been offered the vaccine?”; (**e**) Question: “Have you received the mpox vaccine?”; (**f**) Question: “Why did you choose to get the vaccine?” A = I see myself as being in the risk group; B = I am worried about the consequences of being infected with mpox; C = It was recommended by a health care professional; D = It was recommended by a friend/family member/partner; E = Other; (**g**) Question: “Why did you choose not to get the vaccine?” A = I have already had mpox; B = I am not seeing myself as being in the risk group; C = I do not believe in the efficiency of the vaccine; D = I am using all other sorts of protecting measures in order not to get infected with mpox; E = I have not heard about the vaccine; F = I am allergic; G = Other reason; H = Do not know/do not wish to answer; (**h**) On a scale of 1–5, “how willing are you to get the mpox vaccine?”.

**Table 1 vaccines-11-01167-t001:** Baseline clinical characteristics of the study population.

Clinical Characteristics	All Participants (n = 727)
Age (years)	55.7 (48.3–62.8)
Men	727 (100.0%)
Alcohol (grams per week)	7 (2–14)
Alcohol (more than 10 units per week)	256 (35% [31–39%])
Smoking status	
Ex-smoker	263 (36% [31–39%])
Never smoked	248 (34% [31–38%])
Current smoker	205 (28% [25–32%])
Origin	
Scandinavian	557 (77% [73–80%])
Other European	83 (11% [9–14%])
Other	87 (12% [10–15%])
Plasma HIV RNA < 50 copies/mL	686 (94.4% [92–96%])
Blood CD4^+^ lymphocyte count, cells/µL	695 (260.6)
Education	
No education or a short education	329 (45% [42–49%])
Long education	367 (50% [47–54%])
STD in the preceding two years	184 (25% [22–29%])

STD: sexually transmitted disease.

**Table 2 vaccines-11-01167-t002:** Baseline clinical characteristics for participants stratified by mpox vaccination status.

Clinical Characteristics	Unvaccinated (n = 489)	Vaccinated (n = 238)	*p*-Value	aOR [95% Confidence Interval]
Age (years)	56.9 (48.6–64.6)	53.8 54 (48.0–59.6)	<0.001	0.9 [0.9–1.0]
Alcohol (units per week)	7 (2–15)	6 (1.2–13)	0.3	1.0 [0.9–1.0]
Alcohol (more than 10 unit per week)	175 (36% [32–40%])	81 (34% [28–40%])	0.7	1.0 [0.7–1.4]
Smoking status			0.01	
Ex-smoker	178 (36% [32–41%])	85 (36% [30–42%])		0.9 [0.6–1.4]
Never smoked	153 (31% [27–36%])	95 (40% [34–46%])		1.7 [1.2–2.6]
Current smoker	153 (31% [27–36%])	52 (22% [17–28%])		0.6 [0.4–0.9]
Origin			0.3	
Scandinavian	378 (77% [73–81%])	189 (79% [74–84%])		1.5 [0.9–2.6]
Other European	61 (12% [10–16%])	22 (9% [6–14%])		1.1 [0.5–2.2]
Other	60 (12% [9–16%])	27 (11% [8–16%])		0.7 [0.4–1.2]
Plasma HIV RNA < 50 copies/mL	464 (94% [93–96.6%])	222 (93% [89–96%])	0.5	0.7 [0.4–1.4]
Blood CD4^+^ lymphocyte count, cells/µL	691.3 (253)	702.6 (276.2)		1.0 [1.0–1.0]
Education			0.3	
No education or a short education	227 (46% [42–51%])	102 (43% [36–49%])		0.9 [0.6–1.2]
Long education	239 (49% [44–53%])	128 (54% [47–60%])		Model did not converge
STD in the preceding two years	85 (17% [14–21%])	99 (42% [35–48%])	<0.001	3.1 [2.2–4.6]

STD: sexually transmitted disease.

**Table 3 vaccines-11-01167-t003:** Baseline clinical characteristics for participants who answered the online survey and those who did not.

Clinical Characteristics	Did Not Answer Survey (n = 326)	Answered Survey (n = 401)	*p*-Value
Age (years)	55.3 (47.2–62.9)	56 (49.2–62.6)	0.3
Men	326 (100.0%)	401 (100.0%)	0.2
Alcohol (units per week)	6 (0.0–13.0)	8 (3.0–15.0)	0.03
Alcohol (more than 10 unit per week)	109 (33% [28–39%])	147 (37% [32–42%])	0.4
Smoking status			0.003
Ex-smoker	115 (35% [30–41%])	148 (37% [32–42%])	
Never smoked	93 (29% [24–34%])	155 (39% [34–44%])	
Current smoker	111 (34.0% [29–40%])	111 (28% [23–32%])	
Origin			0.01
Scandinavian	234 (71.8% [30–41%])	323 (81% [76–84%])	
Other European	42 (12.9% [30–41%])	41 (10% [7–14%])	
Other	50 (15.3% [30–41%])	37 (9% [7–12%])	
Plasma HIV RNA < 50 copies/mL	307 (94.2% [92–97%])	379 (95% [92–97%])	0.9
Blood CD4^+^ lymphocyte count, cells/µL	706.4 (270.6)	686.4 (253)	0.4
Education			0.9
No education or a short education	144 (44% [39–50%])	185 (46% [41–51%])	
Long education	162 (50% [44–55%])	205 (51% [46–56%])	
STD in the preceding two years	86 (26% [22–32%])	98 (24% [20–29%])	0.6

STD: sexually transmitted disease.

**Table 4 vaccines-11-01167-t004:** Mpox worry, sexual behavior, and vaccination uptake willingness among MSMWH divided by vaccination status.

Survey Question	Unvaccinated (n = 246)	Vaccinated (n = 155)	*p*-Value
On a scale of 1–5, how worried are you about getting infected with mpox?			<0.001
1	94 (41.2%)	20 (12.9%)	
2	64 (28.1%)	42 (27.1%)	
3	38 (16.7%)	46 (29.7%)	
4	24 (10.5%)	30 (19.4%)	
5	8 (3.5%)	14 (9.0%)	
How many partners have you had in the last 12 weeks?			<0.001
0–1	89 (58.6%)	22 (14.8%)	
1–5	49 (32.2%)	79 (53.0%)	
>5	14 (9.2%)	48 (32.2%)	
Had at least one STD in the preceding two years			<0.001
Yes	33 (14.2%)	60 (38.7%)	
No	199 (85.8%)	95 (61.3%)	
On a scale of 1–5, how willing are you to get the mpox vaccine?	**Been offered a vaccine**	**Not been offered a vaccine**	<0.001
1	10 (6.0%)	35 (18.2%)	
2	5 (3.0%)	15 (7.8%)	
3	4 (2.4%)	28 (14.6%)	
4	12 (7.2%)	31 (16.1%)	
5	132 (79.0%)	74 (38.5%)	

STD: sexually transmitted disease.

## Data Availability

Data are available upon reasonable request to the corresponding author.

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
