# Peer review of "Mpox Incidence and Vaccine Uptake in Men Who Have Sex with Men and Are Living with HIV in Denmark"

_vaccines, 2023, doi:10.3390/vaccines11071167_

Round 1

Reviewer 1 Report

This is a nice descriptive analyses on MPOX incidence and vaccin uptake in  727 MSM with HIV previous recruited in de COCOMO cohort in Denmark. using the  unique datainfrastructure in Denmark enabled the authors to link their cohort to MPOX testing results and vaccin uptake. 

the results are clearly presented and the discussion is balanced and has a clear call for action.  

Minor comments/ suggestions:

- the authors used a < 2 years ago STD as a proxy for risk behaviour. Did they also look at more recent STDs and was syphilis included as an STD (and if so how?) 

- 6/13 cases with MPOX indicated that they having had MPOX was the reason for not being vaccinated. Was their a temporal relation ( ie all cases prior to or just after rollout of vaccination campaigns?)  WOuld be good to clarify this 

Reviewer 2 Report

Dear Authors,

Thanks for your work.

I have found some issues during reviewing your work, hoping to address them.

1.       L 35: experience only few or no symptoms  you can provide your statement with this reference (3390/vaccines10122091)

2.       L 37-38: As of May 23, 2023, 87,529 cases of mpox have been reported globally [3]. … You can mention it case fatality rate.

3.       L 78:  single booster dose…. Which vaccine?

4.       Table 1: eduvation … education

Minor 

Reviewer 3 Report

 The study “Mpox incidence and vaccine uptake in men who have sex with men and are living with HIV” is interesting and well-explained. It provides valuable insights into the incidence of MPox and factors influencing vaccination uptake among MSMWH. However, in my opinion, as the sampling population is chiefly from Copenhagen, Denmark, it would be more appropriate to mention the study area (i.e. Denmark) in the title, as previous studies are available in the literature addressing the similar research question in various countries including USA, China and British Columbia (Canada) etc.

Moreover, I have found that the abstract section does not provide sufficient information on any measures taken to address potential biases resulting from missing data. Understanding the extent and potential impact of missing data is crucial for evaluating the internal validity and reliability of the study. The introduction does not clearly state the significance of investigating the incidence of MPox and the factors associated with vaccine uptake among MSMWH. It would be beneficial to provide a more explicit rationale for why this study is important and how it addresses existing knowledge gaps or research needs.

Furthermore, information about alcohol consumption, smoking, origin, education, and previous sexually transmitted diseases (STDs) was collected through self-reported questionnaires. This method is prone to recall bias and social desirability bias, which may affect the accuracy and reliability of the data. The reliance on self-reported data for important variables raises concerns about the validity of the findings. Additionally, the data collection period for outcomes was relatively short, from May 1, 2022, to October 31, 2022. This short follow-up duration may not capture the long-term incidence of MPox infection or assess the persistence of vaccination effects. More extended follow-up periods would provide a more comprehensive understanding of the outcomes.

Furthermore, the methodology does not mention the response rate of the online survey conducted to collect additional information. The lack of information on the response rate makes it difficult to assess the representativeness of the survey results and the potential for non-response bias. Furthermore, the manuscript does not address the generalizability of the study's findings beyond the specific cohort of mainly well-treated MSMWH. They are considering the potential variations in MPox incidence, vaccination uptake, and associated factors across the MSMWH population which is not well-treated. Authors are also requested to discuss the less-treated MSMWH in this regard.

The results section is well elaborated and well depicted. However, in tables 1-4, I would like to suggest that instead of describing the results in the form of “n(%)”, it would be better to describe it in the form of %(95% C.I of percentage). Moreover, it is also suggested that the adjusted odd ratios described in descriptive results regarding the associated factor should also be incorporated in a separate column in each corresponding table.

Round 2

Reviewer 3 Report

The authors have revised the manuscript accordingly and now it is improved enough to be published.